health and disease and epidemiology

plague, *Yersinia pestis*, climate, British India, third plague pandemic

**Author for correspondence:**
Warren S. D. Tennant
e-mail: Warren.Tennant@warwick.ac.uk

# Climate drivers of plague epidemiology in British India, 1898–1949

Warren S. D. Tennant[1,2], Mike J. Tildesley[1,2,3], Simon E. F. Spencer[1,4] and Matt J. Keeling[1,2,3]

[1]The Zeeman Institute: SBIDER, [2]Mathematics Institute, [3]School of Life Sciences, and [4]Department of Statistics, University of Warwick, Coventry CV4 7AL, UK

 WSDT, 0000-0001-5822-6887; MJT, 0000-0002-6875-7232; SEFS, 0000-0002-8375-5542; MJK, 0000-0003-4639-4765

Plague, caused by *Yersinia pestis* infection, continues to threaten low- and middle-income countries throughout the world. The complex interactions between rodents and fleas with their respective environments challenge our understanding of human plague epidemiology. Historical long-term datasets of reported plague cases offer a unique opportunity to elucidate the effects of climate on plague outbreaks in detail. Here, we analyse monthly plague deaths and climate data from 25 provinces in British India from 1898 to 1949 to generate insights into the influence of temperature, rainfall and humidity on the occurrence, severity and timing of plague outbreaks. We find that moderate relative humidity levels of between 60% and 80% were strongly associated with outbreaks. Using wavelet analysis, we determine that the nationwide spread of plague was driven by changes in humidity, where, on average, a one-month delay in the onset of rising humidity translated into a one-month delay in the timing of plague outbreaks. This work can inform modern spatio-temporal predictive models for the disease and aid in the development of early-warning strategies for the deployment of prophylactic treatments and other control measures.

## 1. Introduction

Plague (caused by infection with *Yersinia pestis*) is a historical bacterial disease which remains a substantial concern to global public health [1]. In the past decade, plague outbreaks have been reported in Madagascar [2], the Democratic Republic of Congo [3] and Peru [4], and the bacterium is regularly detected among different small rodent reservoirs in the USA [5], China [6] and Kazakhstan [7]. These small rodents have the potential to spread the disease to humans via bites from infected fleas [8]. Infected humans will develop tender swollen lymph nodes, known as buboes [9], from which the infection may later spread to the lungs via the bloodstream leading to secondary pneumonic plague [10]. At this point, the bacteria can be directly transmitted from human to human through respiratory droplets [11].

Several pathways within this transmission cycle are influenced by climate [12]. Cold and dry environments can hinder the survival and development rates of flea eggs and larvae [13,14]. Ambient temperatures that are too low or too high can also inhibit flea-gut blockage [15–17]—a proposed mechanism of successful bacterial transmission [18]. However, if blockage has already occurred, the role of temperature on bacterial survival could be more crucial in determining transmission efficiency [19]. The population dynamics and behaviour of rodent hosts are also affected by seasonal variation in temperature and precipitation [20]. High rainfall can flood rodent burrows, driving them towards urban areas [21] and low resources during winter can reduce rodent populations [22]. Combining these factors can have drastic effects on plague epidemiology [23–25]; climate drivers can facilitate the introduction of bacteria into naive

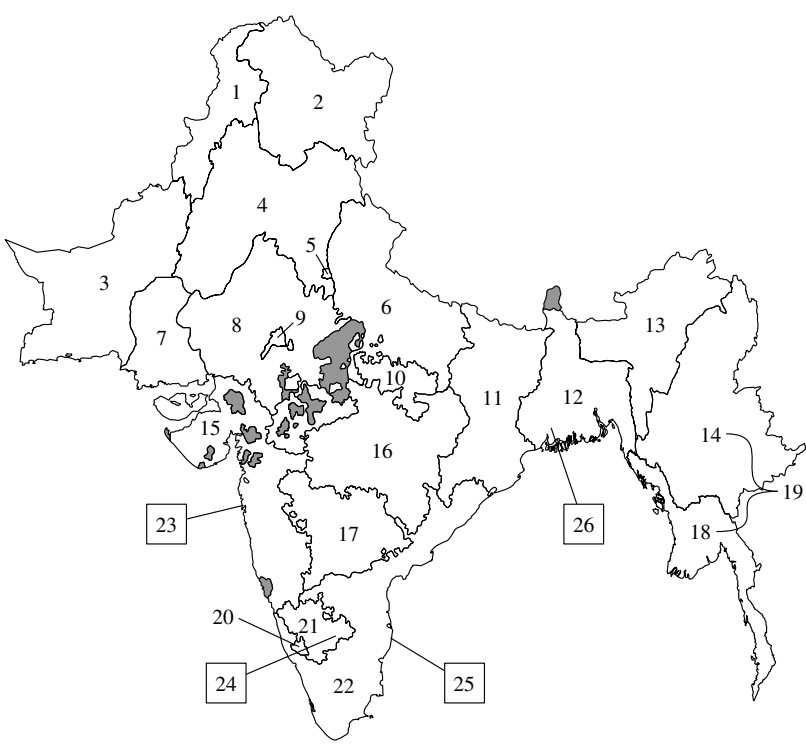

1. North-West Frontier Province
2. Jammu and Kashmir
3. Baluchistan Agency
4. Punjab
5. Delhi
6. United Province of Agra and Oudh
7. Sindh
8. Rajputana Agency
9. Ajmer Merwara
10. Central India Agency
11. Bihar and Orissa
12. Bengal
13. Assam
14. Upper Burma
15. Bombay Presidency
16. Central Province and Berar
17. Hyderabad
18. Lower Burma
19. Burma
20. Coorg
21. Mysore
22. Madras Presidency
23. Bombay City (Mumbai)
24. Bangalore Civil and Military Station
25. Madras City (Chennai)
26. Calcutta City (Kolkata)

**Figure 1.** Provinces in British India. The map shows the location of each province identified in the plague data from the annual Chief Commissioner reports. The data encapsulated modern-day Pakistan, India, Bangladesh and Myanmar. Provinces without any available plague data from 1898 to 1949 are shown in grey.

rodent populations [26,27], from which plague outbreaks can emerge annually within human populations [9,28].

In the absence of an effective licensed vaccine [29], it is important to understand the epidemiological drivers of plague to inform public health planning. Detailed historical records offer a valuable opportunity to study long-term plague on a large spatio-temporal scale given the scarcity of modern-day outbreaks. The extent of the third plague pandemic, originating in Yuhan Province, China around 1855 [30] before spreading globally throughout the nineteenth and twentieth centuries [31], allows us to rigorously quantify the effects of climate on plague epidemiology across different spatial contexts. The effects of climate on the annual cases of plague in China [32] and pre-industrial Europe [33,34] have already been demonstrated; here, we consider the dynamics at a finer scale, and present a 50 year historical dataset of monthly provincial plague-related deaths during the third plague pandemic in British India, one of the most severely affected regions during the third plague pandemic [35,36], including modern-day Pakistan, India, Bangladesh and Myanmar. We analyse how plague emerged annually throughout the region from 1898 to 1949 and, using temperature, rainfall and humidity data from the same time-period, we show the role of climate on the likelihood of outbreaks occurring. Finally, we demonstrate the relationship between the timing of annual plague outbreaks throughout British India against seasonal climate variation.

## 2. Material and methods

### (a) Plague data

After the introduction of plague into British India in 1896, data for monthly plague deaths per province in India were available throughout 1898–1949 in the annual reports of the Chief Sanitary Commissioner of India [36]. The reports also contained monthly plague deaths for Bombay City (Mumbai), Madras City (Chennai), Calcutta City (Kolkata) and Bangalore Civil and Military Station. For Upper and Lower Burma, data were only available from 1904 to 1922; other years contained spatially aggregated averages for the whole of Burma. Provinces constituted parts of modern-day Pakistan, India, Bangladesh and Myanmar. Each location was indexed from the north, starting with North-West Frontier Province, to the southern tip of the Indian Peninsula. The location of each province and corresponding identification number are shown in figure 1.

### (b) Population data

Population data for each province in India were obtained from 1901, 1911, 1921, 1931 and 1941 census of India [37]. Population sizes between these years were estimated using cubic spline interpolation through this data [38]. The number of plague deaths per 1 000 000 individuals per province from 1898 to 1949 was then calculated using these population estimates.

### (c) Climate data

Climate data were obtained from the annual and monthly weather reviews of India provided by the National Oceanic and Atmospheric Administration [39]. Climate records on the same spatio-temporal scale as the plague data were only available for the years between 1907 and 1936. These data contained monthly averages for temperature (in °F), relative humidity and total rainfall (in inches). Monthly weather records were not consistently available for Delhi, Ajmer Merwara, Coorg, Bombay City, Bangalore Civil and Military Station, Madras City and Calcutta. Humidity data were also not available for Upper and Lower Burma, Jammu and Kashmir and Baluchistan Agency. In the place of Lower and

Upper Burma, spatially averaged relative humidity was available for the entirety of Burma. Some records had missing values for monthly climate averages from select provinces. These missing values were replaced by manually averaging across all available data from weather station records found in the appendix of the weather reviews. These climate data were digitized by image processing tables of values in GIMP [40] and performing image recognition in R using the tesseract package [41,42]. In total, 16 837 climate record entries were digitized. All digitized data were then manually checked against the annual and monthly weather reviews and adjusted to match the raw data.

## (d) Historical maps

Historical maps depicting administrative boundaries in the years 1901, 1911, 1921, 1931 and 1941 were obtained from the Administrative Atlas of India [43]. These maps were then digitized in QGIS [44] down to the provincial level. Missing maps (owing to administrative boundary changes in intermediate years) were then constructed. For each year, plague data were then paired with the map which contained the same provinces.

## (e) Timing of plague outbreaks

Wavelet analysis [45] was performed on the time-series of monthly plague-related deaths per 1 000 000 individuals for each province using the R package WaveletComp [46]. This approach deconstructed each time-series into a series of Mortlet wavelets of different periodicities [45]. The periodicity of plague outbreaks in each province was then calculated as the mean period of the wavelet with the largest magnitude across all time points. The timing of annual plague outbreaks and oscillations in temperature, rainfall and humidity was then calculated by comparing the phase of annual wavelet components to give the approximate time during the year at which plague outbreaks and fluctuations in climate peaked for each year. To understand the role of climate on the timing of plague outbreaks, time lags between plague outbreaks and climate for each location, denoted here as $\psi_{t_p/t_c}(t, l)$, were calculated as follows:

$$\psi_{t_p/t_c}(t, l) = \phi_{t_p}(t, l) - \phi_{t_c}(t, l),$$

where $\phi_p(t, l)$ and $\phi_c(t, l)$ are the phases of the annual wavelet components corresponding to time $t$ and location $l$ for the plague outbreaks and climate time-series, $t_p$ and $t_c$, respectively. The mean annual time lag for each province was then calculated by averaging over the time lag across all months in each year.

## (f) Climate effects on plague outbreaks

In order to assess the relationship between climate and plague outbreaks in British India during 1898–1949, regression models were fitted to the empirical data within a Bayesian framework. For all models, weakly informative independent Cauchy priors were used for the coefficients, $\beta_i$, with mean zero and scale parameter equal to 10 and 2.5 for the intercepts and slopes respectively. Exponential priors with rate one were used for variance parameters. For models with multiple predictors, all covariates were orthognalized using QR-decomposition, models fitted and coefficients back-transformed to the scale of the data [47]. All models were fitted using the package rstanarm in R

[48] and convergence and fit was assessed through visual inspection of the posterior predictive distribution and Gelman–Rubin statistic, $\hat{R}$ [49].

Bayes factors [50], $B$, were used to assess the strength of evidence in favour of each model against their respective null model—a model containing no linear covariates. A Bayes factor of greater than one indicated favour towards the model, whereas less than one indicated favour towards the null model. A more detailed interpretation of Bayes factors can be found in the electronic supplementary material, table S1. The R-package bayestestR [51,52] was used to compute Bayes factors via bridge-sampling [53] to estimate the marginal-likelihood of each model (see [54] for more details). Bayes factors were also calculated to determine the relative probability that each model parameter was non-zero, denoted $B_{\beta_i}$, using the Savage-Dickey density ratio [55]. Each model was fitted to temperature, rainfall and humidity data separately. All regression model results are presented in the electronic supplementary material, tables S2–S4.

### (i) Effect of climate on outbreak occurrence

An outbreak was said to occur if the total number of plague-related deaths per 1 000 000 individuals within a year was (strictly) more than some outbreak threshold. The following logistic regression model was then fitted to estimate the probability of outbreaks occurring based on annual climate averages:

$$\chi_\alpha \sim \text{Binom}(1, p),$$

$$\text{logit}(p) = \beta_0 + \beta_1 x + \beta_2 x^2,$$

where $\chi_\alpha$ was whether an outbreak occurred or not given an outbreak threshold $\alpha$, and $x$ denotes annual climate averages. For rainfall, an additional cubic term, $\beta_3 x^3$ was added to the model. Outbreak thresholds of 0, 1, 10 and 100 plague-related deaths per 1 000 000 individuals were tested.

### (ii) Effect of climate on outbreak magnitude

The magnitude of an outbreak was defined as the total number of plague-related deaths reported during each year where an outbreak had occurred given an outbreak threshold of zero. The effects of temperature, rainfall and humidity on outbreak magnitude were then estimated using the following model:

$$\log(y) \sim N(\mu_y, \sigma_y^2),$$

$$\mu_y = \beta_4 + \beta_5 x + \beta_6 x^2,$$

where $y$ was the total number of reported plague deaths per year, $x$ denotes annual climate averages and $\sigma_y^2$ is the variance of $\log(y)$ to be estimated. Similarly to the above regression model, an additional cubic term, $\beta_7 x^3$ was added to the model for rainfall.

### (iii) Effect of climate on outbreak timing

In order to estimate the relationship between the timing of plague outbreaks and oscillations in climate, the following model was fitted to the time at which plague outbreaks peaked during the year within each province, $\tau_p$:

$$\tau_p \sim N(\mu_{\tau_p}, \sigma_{\tau_p}^2),$$

$$\mu_{\tau_p} = \beta_8 + \beta_9 \tau_c,$$

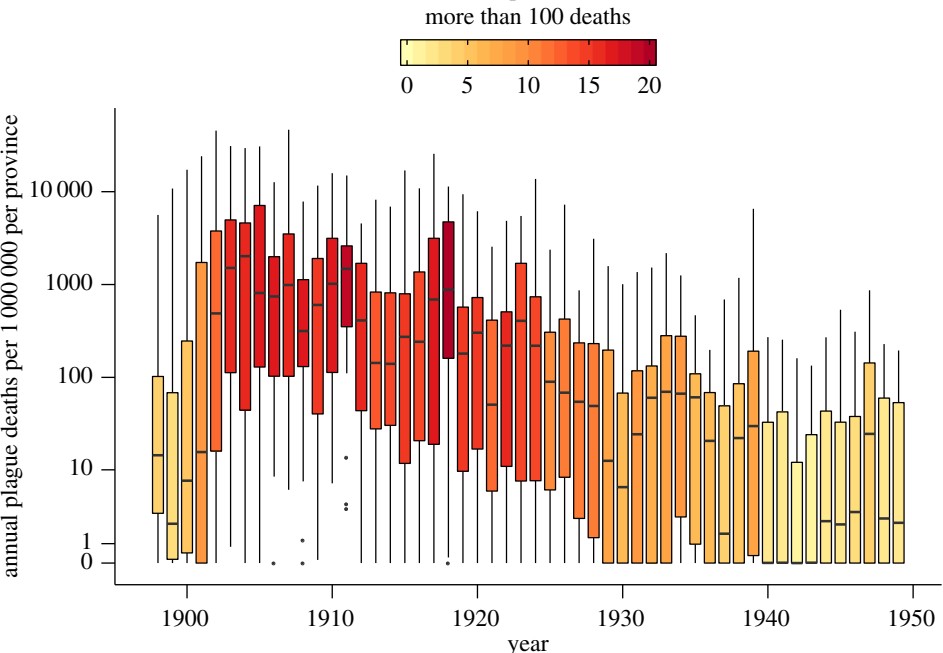

**Figure 2.** Annual plague deaths in each province of British India from 1898 to 1949. The total number of plague deaths showed large variation between each province within the same year. When the third pandemic began in around 1898, the total number of plague-related deaths per province rapidly increased each year. From 1905 onwards, plague-related deaths steadily decreased to low levels during the 1940s. (Online version in colour.)

where $\tau_c$ denotes the time at which oscillations in climate peak (calculated from Wavelet analysis), and $\sigma^2_{\tau_p}$ is the variance of $\tau_p$ to be estimated. The estimated coefficient $\beta_9$ represented the additional (on average) time-lag of a plague outbreak given a one-month time-lag in oscillations of each climate variable.

## 3. Results

Plague was first reported to the Chief Sanitary Commissioner of India during the latter part of 1896 in the west of the Indian Peninsula. By 1898, plague-related deaths were reported throughout British India. Annual deaths within each province increased until 1905, when over 22 of the 25 provinces experienced over 100 deaths per one million individuals (figure 2). The size of annual outbreaks in each province then decreased until 1930 to around five deaths per one million individuals. From there on, a low level of background transmission was maintained until 1950. Over 13 million plague-related deaths in total were reported across British India from 1898 to 1949.

### (a) Spatial spread of plague in British India from 1898 to 1949

During this period, the north, northwest and west of the Indian Peninsula were the most severely affected. These regions included Bombay Presidency and Punjab provinces, which reported over two and three million cases, respectively. By contrast, cases were infrequently reported in the drier far northwest and wetter northeast of British India, including the provinces Baluchistan Agency, North-West Frontier Province and Assam. This high degree of spatial variation of cases was largely consistent between years (electronic supplementary material, figure S1). That is, provinces

which had experienced large outbreaks in previous years reported a high number of cases in subsequent years.

To robustly demonstrate these trends, wavelet analysis was performed on the monthly reported cases within each province. By comparing the annual components of each wavelet, the timing of each outbreak during the year was calculated. This analysis confirmed the annual frequency of outbreaks within each province, and also showed that the timing of these outbreaks was largely consistent over time (electronic supplementary material, figures S2–S4). However, outbreak timing varied considerably between provinces (figure 3). Outbreaks started in the south of the Indian Peninsula in October each year and radiated out towards the north over a period of six to seven months. Simultaneously, annual plague outbreaks cycled between the Burmese native Shan state in September, to Upper Burma in January, finishing in Lower Burma in April. We also found that the timing of outbreaks between most dense population centres (Bangalore Civil and Military Station, Madras City and Calcutta City) and their respective provinces was within one month of one another. By contrast, the timing of outbreaks between Bombay City (Mumbai) and Bombay Presidency differed by approximately six months (electronic supplementary material, figure S5).

### (b) Effect of humidity on outbreak occurrence

We tested the probability of annual outbreaks occurring based on observed climate values for each province, using different thresholds to define an outbreak. A range of outbreak thresholds was tested, from more than 0 deaths per 1 000 000 individuals to 100 deaths per 1 000 000 individuals within a calendar year. For all thresholds tested ($B > 1000$), moderate mean annual humidity levels of between 60% and 80% were associated with plague outbreaks (figure 4). Outbreaks were 1.9 (95% credible interval

Proc. R. Soc. B **287**: 20200538

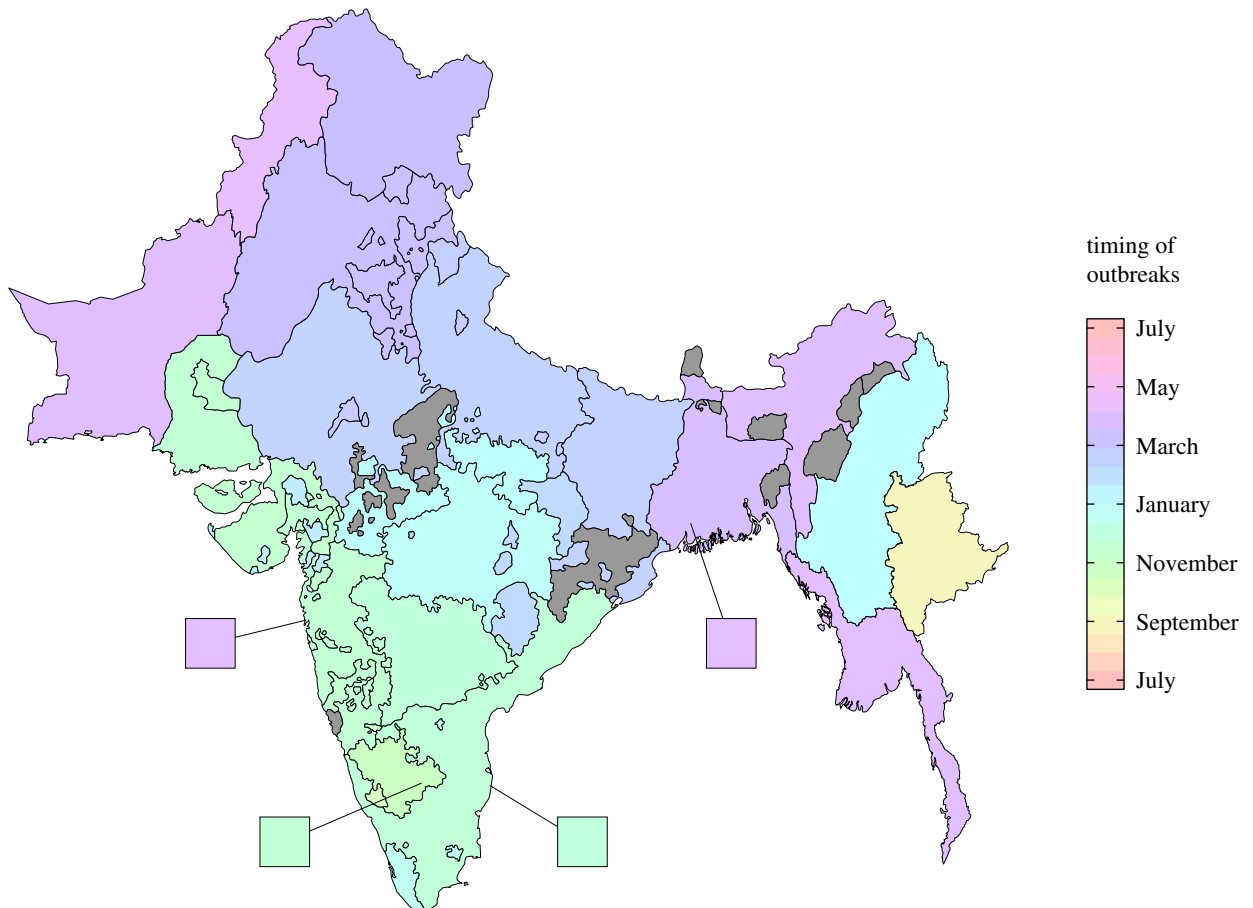

**Figure 3.** Timing of plague outbreaks in British India from 1898 to 1949. The mean timing of outbreaks in each province of British India from 1898 to 1949. Outbreaks began in the south of the Indian Peninsula and cascaded out towards the north. At the same time, plague outbreaks cycled around the Burmese Shan state and Burma. The timing of outbreaks was calculated by comparing an annual sinusoidal function with the annual component of plague incidence determined from wavelet decomposition. Grey provinces indicate insufficient plague data to calculate outbreak timing. (Online version in colour.)

(CI) = [1.5, 2.6]) and 2.2 (95% CI = [1.7, 2.8]) times more likely to occur given moderate annual humidity levels (60–80%) compared to lower (40–60%) or higher (80–100%) humidity. Moderate humidity levels were also associated with outbreaks of greater magnitude, although there was substantial variation in outbreak magnitude across all observed mean annual humidity values (electronic supplementary material, figure S6). Outbreaks were also more likely to occur at extreme temperatures and moderately low precipitation ($B > 1000$), however, this relationship was generally not consistent across different outbreak thresholds (electronic supplementary material, figures S7–S8). From herein, outbreaks were defined as more than 10 per 1 000 000 plague-related deaths in a single year.

## (c) Effect of humidity on outbreak timing

We investigated if the systematic pattern (south to north) in the timing of plague outbreaks could be explained by annual oscillations in climate. To that end, temperature, rainfall and humidity data for each province during 1907–1936 was obtained from written records (electronic supplementary material, figure S9 shows these data over time in five different provinces throughout British India). The annual periodicity of the climate data was confirmed using wavelet analysis, and wavelets for each climate variable were compared against annual wavelet components generated from the plague outbreak data. This allowed us to quantify the lag

in time between oscillations in climate and plague outbreaks for each year and location.

We found that the time delays between oscillations in climate and plague outbreaks were largely consistent across time within each province. That is, within the same region, there was little variation in the lag between outbreaks (when they occurred) and temperature, rainfall and humidity (electronic supplementary material, figure S10). However, these time lags were different between locations and exhibited a similar systematic spatial pattern to the plague outbreak data: shorter time lags were found in the southern tip of the Indian Peninsula and lengthened towards the north of India and modern-day Pakistan. In general, over half of all plague outbreaks lagged four to eight months behind peak rainfall and six to nine months behind peak temperature. The time lags between oscillations in humidity and plague outbreaks were the shortest and most similar across space with delays of three to five months between the peak of humidity and the peak of plague deaths (electronic supplementary material, figure S11).

Owing to the variation in lag times between provinces, we investigated if the timing of seasonal oscillations in climate could reliably infer outbreak timing. Across all of British India and all years, oscillations in temperature and rainfall had strong relationships with plague outbreaks (Spearman $\rho_S = 0.75$, Pearson $\rho_P = 0.64$, $B > 1000$ and $\rho_S = -0.62$, $\rho_P = -0.57$, $B > 1000$, respectively) (electronic supplementary material, figure S12). A simpler linear relationship was

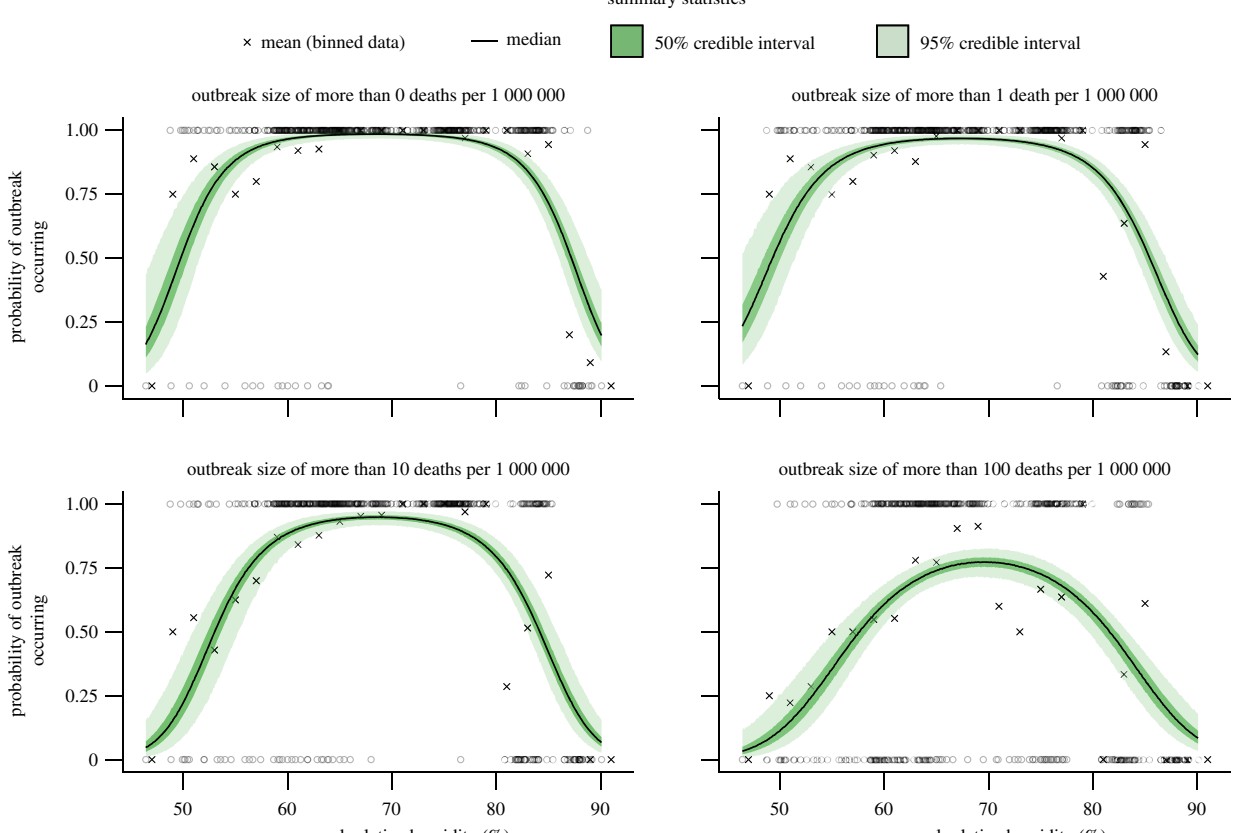

**Figure 4.** Humidity effects on outbreak occurrence. Outbreaks had a higher probability of occurring at moderate levels of mean annual humidity. At low and high levels of humidity, outbreaks of any size rarely occurred, $B > 1000$. Credible intervals were calculated from fitting the mean humidity per year per province to whether outbreaks occurred or not under different thresholds for defining an outbreak. Circles denote the binary data that was fitted, and crosses show the mean probability of an outbreak occurring at binned humidity values. A quadratic polynomial was fitted within a Bayesian framework with weakly informative Cauchy priors on coefficients and assumed binomial error structure using the `rstanarm` package in R. (Online version in colour.)

detected ($\rho_S = 0.53$, $\rho_P = 0.53$, $B > 1000$) between seasonal changes in humidity and the timing of outbreaks (figure 5). At the national scale, outbreaks would occur approximately one month (95% CI = [0.9, 1.2]) later on average throughout British India for every month peak humidity was delayed.

Focusing on the variation within a province, we again find that there is a linear relationship between the timing of outbreaks and humidity oscillations ($\rho_S = 0.43$, $\rho_P = 0.42$, $B > 1000$) but no evidence to support a relationship with temperature or rainfall ($B < 0.001$ and $B < 0.001$, respectively). In particular, within a province we find that a one-month delay to oscillations in humidity translated to a three and a half-week delay in the timing of outbreaks on average (95% CI = [2.5, 4.5]), although there was considerable variability—from two weeks in Madras Presidency to 12 weeks in Burma (electronic supplementary material, figures S13–S15).

## 4. Discussion

Zoonotic pathways of *Y. pestis* transmission are affected by climate, driving plague epidemiology. Optimal environmental factors, such as temperature, precipitation and humidity, can create the ideal conditions from which plague can emerge from zoonotic rodent reservoirs. The exact relationship between climate and plague outbreaks varies between different spatio-temporal contexts and is typically inferred from short time periods. Therefore, in order to understand these relationships in more detail, analyses of

longitudinal data across spatial foci are important. Given the global extent of the third plague pandemic [31,56], historical datasets from this period allow us to rigorously explore these effects on large spatio-temporal scales. During the course of the third plague pandemic, India reported over 13 million plague-associated deaths [35,57]. British India was, therefore, an appropriate setting for studying the effects of climate on plague epidemiology. To that end, we were interested in whether climate could explain the occurrence, severity and timing of outbreaks in British India during the third plague pandemic.

We started by analysing the timing of plague outbreaks in British India from 1898 to 1949. Similar to the subset of data presented by Yu & Christakos [58], outbreaks would begin annually in the southern tip of the Indian Peninsula and spread up towards the north over six months. Outbreaks would simultaneously start in the east of Burma, and move towards the south. These two separate cycles of plague spread have previously been demonstrated through phylocartographic analysis of *Y. pestis* isolates from the third pandemic, where strains in Burma were found to be distinct from the those of the Indian Peninsula [31]. Their study inferred different directions of plague spread, however. That is, the data presented in this study implied that Burmese plague originated in the north and spread southward, whereas they found that strains were introduced from Vietnam and spread north towards China. Plague was also introduced into Bombay and Calcutta City via shipping routes during the late nineteenth century [59,60], but annual outbreaks would not start in these regions.

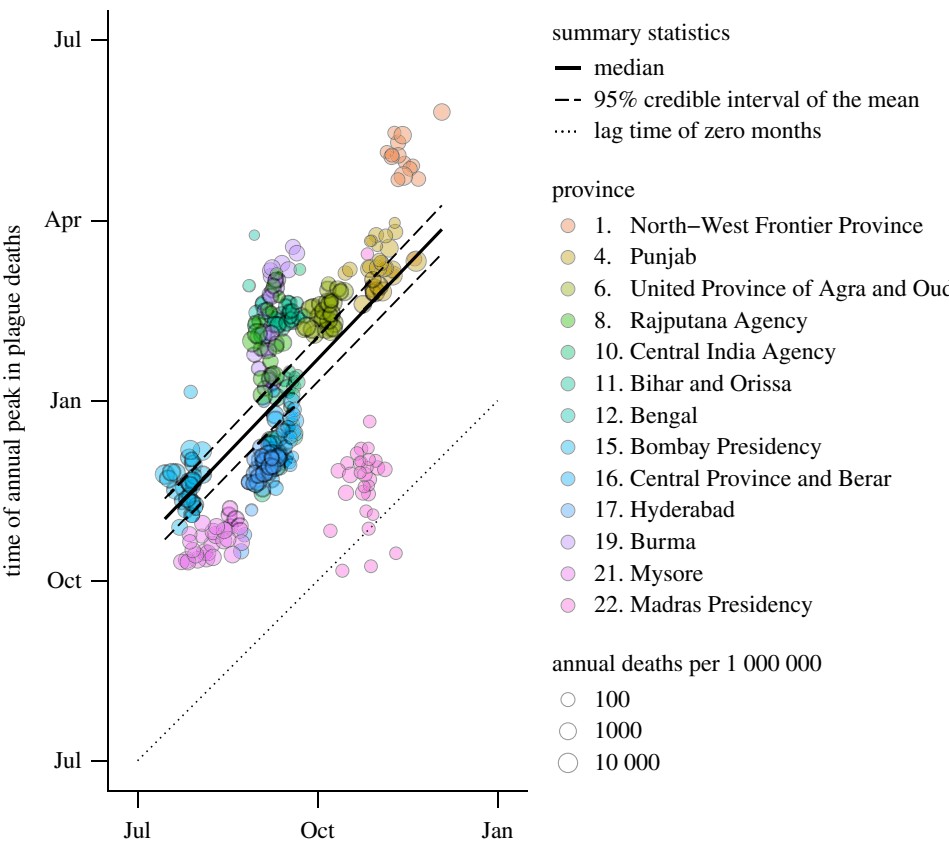

**Figure 5.** Oscillations in humidity were correlated with outbreaks. The timing of seasonal changes in humidity had a strong linear relationship with the timing of plague outbreaks, $B > 1000$. Later peaks in humidity were associated with later occurrence of plague outbreaks at the national level. Outbreaks were defined as more than 10 plague-related deaths per one million individuals within a single year and province. (Online version in colour.)

These contrasting findings may imply that once plague has been introduced and established among zoonotic reservoirs, the re-emergence of outbreaks is dictated by local environmental and socio-demographic factors. Indeed, *Rattus rattus* (formerly *Mus rattus*) and *Rattus norvegicus* (formerly *Mus decumanus*) from several regions of British India tested positive for plague in the early twentieth century [61]. However, if local conditions are insufficient to allow the establishment of zoonotic reservoirs, then large scale outbreaks might rarely occur. This was the case in Assam in the north east of India, where exceptionally wet climates throughout the year and poor transport links with neighbouring provinces [62,63], crucial in the spread of plague throughout pre-industrial Europe [64,65], hindered transmission.

Outbreaks of plague were more likely to occur at moderate relative humidity levels of between 60% and 80% than at higher or lower humidity, which was independent of how outbreaks were defined in terms of reported deaths. This was in agreement with a similar study looking at the effects of climate during the third pandemic in China [32,66], finding that plague spread fastest at moderate levels of wetness. These findings may be down to the sensitivity of flea egg and larvae survival rates to changes in soil moisture, a proxy for humidity. This could explain the substantial difference in average outbreak timing between Bombay City (Mumbai) and Bombay Presidency. Despite the absence of Bombay City in the presented climate data, Bombay City did indeed have generally higher humidity than the rest of the region encapsulating Bombay Presidency [67]. In contrast to the rest of the country, Bombay City may have, therefore,

only been climatically suitable for plague transmission once humidity dropped, offsetting outbreak timing by roughly six months. This feature could be an important consideration when investigating the effects of climate on modern outbreaks, as regions such as Madagascar and the Democratic Republic of Congo have very high humidity throughout the year. The outbreaks in Madagascar in fact do correlate with time periods where humidity subsides, and the suitability of local flea species improves drastically [68].

The timing of plague outbreaks in British India was associated with seasonal changes in humidity. We found that a one month delay in humidity led to an approximate one month time delay in the plague outbreak on average, but with substantial variation between provinces. In addition to driving change in flea suitability, oscillations in humidity may indirectly influence rodent population dynamics via timing of harvest. Harvest has been suggested to draw rodents towards rural agriculture [69], putting farm workers at increased risk of infection. In the case of British India, this may explain the bi-annual nature of plague outbreaks within some regions, such as Burma, where harvest used to occur twice a year [70]. Agricultural land-use has indeed previously been correlated with higher seroprevalence of *Y. pestis* among rodents compared with other land types [71], but it is unclear where they were infected. That is, either rodents became infected before migrating to agricultural land or were infected from fleas within the fields themselves. Regardless, once harvest is complete, food trade into urban centres could drive infected rat populations into urban areas, sparking a large outbreak.

Given our findings on the effects of humidity on the occurrence and timing of plague outbreaks, one might expect there to be equally strong associations between temperature, rainfall and plague. We did not find this in our data. This was in contrast to a study by Xu et al. [27], demonstrating high speeds of plague spread globally during the third plague pandemic influenced by temperature. Our findings may be attributable to temperature and rainfall gradients being fairly flat across most of the Indian Peninsula (in terms of magnitude and timing), so it was unlikely that temperature or rainfall played decisive roles in dictating plague outbreaks in British India during the third pandemic. On the other hand, there was evidence to suggest that rainfall was positively associated with the timing of outbreaks in Burma. This lack of consistent and uniform drivers in our data was unsurprising given previous claims that plague outbreaks can occur under a diverse set of landscapes for a wide range of environmental conditions [72].

The statistical analysis of epidemiological data is fraught with potential limitations. Most notability data quality may be highly variable in quality with discrepancies in both space and time [73]. This is probably an issue with our data, where more populated regions may have had greater resources for observing and recording plague cases. Cases may also have been over-reported where deaths were misdiagnosed as being owing to plague. This is of particular concern in the case of modern-day pneumonic plague outbreaks which can be misdiagnosed in conjunction with other respiratory infections [74]. Pneumonic plague transmission could add additional layers of complexity in trying to understand the drivers of plague epidemiology, as once an outbreak is established, both rodent and flea populations are no longer required. However, the scale of under- or over-reporting over both time and space should not heavily influence our findings, which are primarily concerned with the occurrence and seasonal timing of outbreaks. We stress that climate alone is not enough to infer the size of an outbreak and we also restricted our analyses to relatively large scale outbreaks, eliminating concern about including introduction events which may bias results.

Historical records were only available at the provincial level for the monthly plague deaths and climate data. The space–time aggregation of these data across a wide geographical and temporal scale may lead to inappropriate generalizations of climate effects at lower resolutions. That is, the mechanisms that underlie plague epidemiology are complex and poorly understood, such that other abiotic factors may dictate the timing of plague outbreaks over finer spatial scales, and thus the statistical approach taken here may only be applicable within spatial contexts similar to our data. This does raise an area for future work where the influence of climate and socio-ecological factors, such as rat population dynamics, on plague epidemiology should be studied at the sub-provincial level. Altitude and the cold, dry season have indeed been correlated with plague foci in Madagascar [2,75]. Given its climatic diversity [76], perhaps temperature, precipitation and humidity could explain the spatially heterogeneous patterns of plague found in Madagascar [77]. However, without a clear mechanistic understanding linking climate factors to plague epidemiology, the direct application of our results to modern outbreak settings might be inappropriate.

In summary, we have shown that humidity was the most important climate factor in dictating the occurrence and timing of plague outbreaks during the third plague pandemic in British India. Humidity data could therefore go a long way in assessing disease risk using spatio-temporal prediction models from recent bubonic plague outbreaks [78]. However, it would also be important to understand how human social contact networks influence plague spread, in order to minimize the impact of pneumonic plague outbreaks [79]. By monitoring local climate factors, our findings could enhance the identification of regions with imminent outbreak risk, thus improving the management of chemoprophylaxis and highlighting settings in which imminent candidate vaccines could be effectively trialled.

Data accessibility. All data and code are made available through the GitHub repository: wtennant/plague_india [80].

Authors' contributions. All authors contributed to the writing and revision of the manuscript.

Competing interests. We declare we have no competing interests.

Funding. This research is funded by the Department of Health and Social Care using UK Aid funding and is managed by the National Institute for Health Research (NIHR) (Global Health Research (PR-OD-1017-20002) to M.J.K., M.J.T. and S.E.F.S.). The views expressed in this publication are those of the authors and not necessarily those of the Department of Health and Social Care. S.E.F.S. also greatly acknowledges support from the Medical Research Council (MR/P026400/1).

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
