## [Reviewer comments · Proceedings of the Royal Society B: Biological Sciences]

Review History

RSPB-2020-0538.R0 (Original submission)

Review form: Reviewer 1

Recommendation

Accept with minor revision (please list in comments)

Scientific importance: Is the manuscript an original and important contribution to its field?

Good

General interest: Is the paper of sufficient general interest?

Good

Quality of the paper: Is the overall quality of the paper suitable?

Good

Is the length of the paper justified?

Yes

Should the paper be seen by a specialist statistical reviewer?

No

Do you have any concerns about statistical analyses in this paper? If so, please specify them explicitly in your report.

No

It is a condition of publication that authors make their supporting data, code and materials available - either as supplementary material or hosted in an external repository. Please rate, if applicable, the supporting data on the following criteria.

Is it accessible?

Yes

Is it clear?

Yes

Is it adequate?

Yes

Do you have any ethical concerns with this paper?

No

Comments to the Author

The manuscript entitled "Climate drivers of plague epidemiology in British India, 1898-1949", which is based on the monthly plague deaths and climate data from 25 provinces in British India between 1898 and 1949 to investigate the influence of temperature, rainfall, and humidity on the occurrence, severity, and timing of plague outbreaks. The manuscript is well-written, the methods are sound, and the results are well-presented. Therefore, I would recommend this manuscript be accepted for publication after minor revision.

Here are my suggestions for the manuscript:

Study area: Perhaps the study area is determined by the availability of the historical plague records. But, it will be good if the authors could further illustrate the importance of British India in examining the plague dynamics.

Study period: I think the authors have implicitly explained why the study period is delimited as 1898-1949 in lines 116-122. But, it will be good if the authors could state it clearly in the earlier sections in the manuscript.

Plague reservoir: Any plague reservoir in British India? The presence of the plague reservoir may mediate the connection between climatic forcings and plague outbreaks.

Pre-industrial period: Apart from historical China, there are also some studies examining the influence of climatic forcings on plague outbreaks in pre-industrial Europe, which may be relevant to this study:

Yue, R.P.H. and Lee, H.F. (2020) Drought-induced spatio-temporal synchrony of plague outbreak in Europe. *Science of The Total Environment* 698: 134138.

Yue, R.P.H. and Lee, H.F. (2018) Pre-industrial plague transmission is mediated by the synergistic effect of temperature and aridity index. *BMC Infectious Diseases* 18: 134.

Line 285-286: For the human social contact networks, they may consist of transportation and trading networks. The following references may be relevant:

Yue, R.P.H., Lee, H.F., and Wu, C.Y.H. (2017) Trade routes and plague transmission in pre-industrial Europe. *Scientific Reports* 7: 12973.

Yue, R.P.H., Lee, H.F., and Wu, C.Y.H. (2016) Navigable rivers facilitated the spread and recurrence of plague in pre-industrial Europe. *Scientific Reports* 6: 34867.

Line 287-289: "These findings could then be used to manage the deployment of chemoprophylaxis and assess imminent vaccination clinical trial designs in real outbreak scenarios." I feel sorry to say that I have difficulty to link the findings in the study with the above implication. Please further specify how the findings could serve this purpose.

Review form: Reviewer 2

Recommendation

Accept with minor revision (please list in comments)

Scientific importance: Is the manuscript an original and important contribution to its field?

Good

General interest: Is the paper of sufficient general interest?

Good

Quality of the paper: Is the overall quality of the paper suitable?

Excellent

Is the length of the paper justified?

Yes

Should the paper be seen by a specialist statistical reviewer?

No

Do you have any concerns about statistical analyses in this paper? If so, please specify them explicitly in your report.

No

It is a condition of publication that authors make their supporting data, code and materials available - either as supplementary material or hosted in an external repository. Please rate, if applicable, the supporting data on the following criteria.

Is it accessible?

Yes

Is it clear?

Yes

Is it adequate?

Yes

Do you have any ethical concerns with this paper?

No

Comments to the Author

Comments in attached file. (See Appendix A)

Decision letter (RSPB-2020-0538.R0)

27-Apr-2020

Dear Dr Tennant:

Your manuscript has now been peer reviewed and the reviews have been assessed by an Associate Editor. The reviewers' comments (not including confidential comments to the Editor) and the comments from the Associate Editor are included at the end of this email for your reference. As you will see, the reviewers have raised some issues with your manuscript and we would like to invite you to revise your manuscript to address them.

Research ethics:

Use of animals and field studies:

Please submit a copy of your revised paper within three weeks. If we do not hear from you within this time your manuscript will be rejected. If you are unable to meet this deadline please let us know as soon as possible, as we may be able to grant a short extension.

Best wishes,
Professor Hans Heesterbeek
mailto: proceedingsb@royalsociety.org

Associate Editor

Board Member: 1

Comments to Author:

Please consider carefully the revisions suggested by the two reviewers, paying particular attention to the issues raised concerning the presentation of the statistical analyses and the limitations to the interpretation of mechanistic understanding based on the statistical approaches used.

Reviewer(s)' Comments to Author:

Referee: 1

Comments to the Author(s)

The manuscript entitled "Climate drivers of plague epidemiology in British India, 1898-1949", which is based on the monthly plague deaths and climate data from 25 provinces in British India between 1898 and 1949 to investigate the influence of temperature, rainfall, and humidity on the occurrence, severity, and timing of plague outbreaks. The manuscript is well-written, the methods are sound, and the results are well-presented. Therefore, I would recommend this manuscript be accepted for publication after minor revision.

Here are my suggestions for the manuscript:

Study area: Perhaps the study area is determined by the availability of the historical plague records. But, it will be good if the authors could further illustrate the importance of British India in examining the plague dynamics.

Study period: I think the authors have implicitly explained why the study period is delimited as 1898-1949 in lines 116-122. But, it will be good if the authors could state it clearly in the earlier sections in the manuscript.

Plague reservoir: Any plague reservoir in British India? The presence of the plague reservoir may mediate the connection between climatic forcings and plague outbreaks.

Pre-industrial period: Apart from historical China, there are also some studies examining the influence of climatic forcings on plague outbreaks in pre-industrial Europe, which may be relevant to this study:

Yue, R.P.H. and Lee, H.F. (2020) Drought-induced spatio-temporal synchrony of plague outbreak in Europe. *Science of The Total Environment* 698: 134138.

Yue, R.P.H. and Lee, H.F. (2018) Pre-industrial plague transmission is mediated by the synergistic effect of temperature and aridity index. *BMC Infectious Diseases* 18: 134.

Line 285-286: For the human social contact networks, they may consist of transportation and trading networks. The following references may be relevant:

Yue, R.P.H., Lee, H.F., and Wu, C.Y.H. (2017) Trade routes and plague transmission in pre-industrial Europe. *Scientific Reports* 7: 12973.

Yue, R.P.H., Lee, H.F., and Wu, C.Y.H. (2016) Navigable rivers facilitated the spread and recurrence of plague in pre-industrial Europe. *Scientific Reports* 6: 34867.

Line 287-289: "These findings could then be used to manage the deployment of chemoprophylaxis and assess imminent vaccination clinical trial designs in real outbreak scenarios." I feel sorry to say that I have difficulty to link the findings in the study with the above implication. Please further specify how the findings could serve this purpose.

Referee: 2

Comments to the Author(s)

Comments in attached file.

Author's Response to Decision Letter for (RSPB-2020-0538.R0)

See Appendix B.

RSPB-2020-0538.R1 (Revision)

Review form: Reviewer 1

Recommendation

Accept as is

Scientific importance: Is the manuscript an original and important contribution to its field?
Good

General interest: Is the paper of sufficient general interest?
Good

Quality of the paper: Is the overall quality of the paper suitable?
Excellent

Is the length of the paper justified?
Yes

Should the paper be seen by a specialist statistical reviewer?
No

Do you have any concerns about statistical analyses in this paper? If so, please specify them explicitly in your report.
No

It is a condition of publication that authors make their supporting data, code and materials available - either as supplementary material or hosted in an external repository. Please rate, if applicable, the supporting data on the following criteria.

Is it accessible?
Yes

Is it clear?
Yes

Is it adequate?
Yes

Do you have any ethical concerns with this paper?
No

Comments to the Author

The authors have addressed all of my concerns. I recommend the paper be accepted for publication.

Decision letter (RSPB-2020-0538.R1)

20-May-2020

Dear Dr Tennant

I am pleased to inform you that your (excellent and interesting) manuscript entitled "Climate drivers of plague epidemiology in British India, 1898–1949" has been accepted for publication in Proceedings B.

Open Access

Paper charges

Sincerely,

Professor Hans Heesterbeek

Associate Editor:

Board Member: 1

Comments to Author:

(There are no comments.)

Board Member: 2

Comments to Author:

(There are no comments.)

Appendix A

The authors use a novel historical dataset of plague data from British India, coupled with historic climate data, to look at the effect of climate on plague incidence. The authors find that relative humidity appears to be correlated with the size and timing of plague cases, but not temperature or precipitation. I find the paper to be very thorough and well-presented (exceptional figures!), and the main results convincing. I believe the paper should be published in Proc. B, however, I have a few comments on the statistical analysis.

Comments:

I think statistical results could be better presented in a few places. The authors appear not to report p-values in the text but do mention significance. I think including p-values would help guide the reader. Particularly in Line 184/185 when discussing significant RH results versus T and P? Also figures S6/S11/S12 should have p-values in the captions.

Related line 106/108/109, I like how the models are laid out clearly in the methods. I think it would be nice, for reference, to have a table in the supplement showing the results of each of these three regressions including p-values for the 3 different climate variables, showing with and without controls for province (i.e. removing mean province outbreaks size as is done in line 183).

Did the authors try to control for common temporal trends across locations? This is often done with year dummies e.g. in a classic panel regression framework where both spatial and temporal controls are included?

Fig4, for lower end of humidity range ~ 40 , e.g. in the top two plots, there seems to be no observations (little circles). Is the quadratic only fitting to the upper zero points? i.e. is it really a quadratic fit or would a gam/spline etc work better? Or are there points here that are not visible?

Fig4, I think the temperature and precipitation version of this plot should be shown in the supplement, even if not significant.

Line 105. Did you consider combined models e.g. temperature and precipitation together? i.e. $\beta_1 * T + \beta_2 * T^2 + \beta_3 * P + \beta_4 * P^2$. I imagine it might have similar explanatory power to RH.

I think the authors need to discuss the lack of mechanistic understanding as a caveat. I appreciate that the authors cannot easily unpack mechanisms, and the plague system is particularly complex. However, given the authors are only using statistical approaches and not using a mechanistic model, the limitations of this approach, in terms of lack of mechanistic understanding, should be discussed. Without a clear understanding of the mechanism linking RH to plague incidence, it is hard to say whether the approximation will hold in other countries still experiencing plague outbreaks.

Appendix B

Response to Referees

Associate editor

‘Please consider carefully the revisions suggested by the two reviewers, paying particular attention to the issues raised concerning the presentation of the statistical analyses and the limitations to the interpretation of mechanistic understanding based on the statistical approaches used.’

We have addressed the suggestions of the two reviewers below. In particular, we have improved the presentation of the statistical analyses by including results of our models throughout the main text and in supplementary tables. We have also added discussion points on the limitations of the statistical approaches used with reference to the lack of understanding on the mechanisms that link climate and plague epidemiology. The manuscript with tracked changes is attached to this document following responses to each reviewer.

Reviewer #1

‘Study area: Perhaps the study area is determined by the availability of the historical plague records. But, it will be good if the authors could further illustrate the importance of British India in examining the plague dynamics.’

Thank-you for your comment. British India was one of the most severely affected regions in the third plague pandemic with over 13 million plague-related deaths. This is why we chose to study this region. We have included this within our introduction.

‘Study period: I think the authors have implicitly explained why the study period is delimited as 1898-1949 in lines 116-122. But, it will be good if the authors could state it clearly in the earlier sections in the manuscript.’

Plague records for British India were indeed restricted to 1898–1949. We have now stated this earlier in our manuscript, in the introduction and methodology.

‘Plague reservoir: Any plague reservoir in British India? The presence of the plague reservoir may mediate the connection between climatic forcings and plague outbreaks.’

In the six reports on Plague Investigations in [British] India, the authors consistently noted that plague-infected *Rattus rattus* (formerly *Mus rattus*) and *Rattus norvegicus* (formerly

Mus decumanus) were found in certain provinces of British India in the early 20th century. We have now made reference to these reports in our manuscript.

‘Pre-industrial period: Apart from historical China, there are also some studies examining the influence of climatic forcings on plague outbreaks in pre-industrial Europe, which may be relevant to this study:

Yue, R.P.H. and Lee, H.F. (2020) Drought-induced spatio-temporal synchrony of plague outbreak in Europe. Science of The Total Environment 698: 134138.

Yue, R.P.H. and Lee, H.F. (2018) Pre-industrial plague transmission is mediated by the synergistic effect of temperature and aridity index. BMC Infectious Diseases 18: 134.’

Thank-you for bringing these articles to our attention. We have included these as references within our manuscript.

‘Line 285-286: For the human social contact networks, they may consist of transportation and trading networks. The following references may be relevant:

Yue, R.P.H., Lee, H.F., and Wu, C.Y.H. (2017) Trade routes and plague transmission in pre-industrial Europe. Scientific Reports 7: 12973.

Yue, R.P.H., Lee, H.F., and Wu, C.Y.H. (2016) Navigable rivers facilitated the spread and recurrence of plague in pre-industrial Europe. Scientific Reports 6: 34867.’

Thank-you for also bringing these articles to our attention. We have included these as citations within our discussion.

‘Line 287-289: “These findings could then be used to manage the deployment of chemoprophylaxis and assess imminent vaccination clinical trial designs in real outbreak scenarios.” I feel sorry to say that I have difficulty to link the findings in the study with the above implication. Please further specify how the findings could serve this purpose.’

Our work identified humidity as a key covariate of plague’s epidemiological dynamics. By monitoring humidity at a spatio-temporal scale, it could be used (alongside other covariates) to assess imminent outbreak risk. This would then allow response teams to appropriately allocate chemoprophylaxis to areas of need and identify times and locations where vaccines could be trialled effectively. We’ve added these additional details on how our findings could serve these purposes.

Reviewer #2

'I think statistical results could be better presented in a few places. The authors appear not to report p-values in the text but do mention significance. I think including p-values would help guide the reader. Particularly in Line 184/185 when discussing significant RH results versus T and P? Also figures S6/S11/S12 should have p-values in the captions.'

Thank-you for highlighting an important point. We realise the p-values will be more familiar to many readers, but they are not consistent with the Bayesian approaches used throughout this manuscript. We therefore use Bayes factors, analogous to p-values, to assess the strength of evidence in support of each model. We also used Bayes factors to assess the strength of evidence that each linear predictor, β_i , is non-zero. We now state this clearly within our methodology. We have adjusted lines 184/185 to more clearly reflect this choice. In order to better guide the reader, we have included the Bayes factors in the main text and figure captions, in addition to the median, and 95% credible interval of each linear predictor β_i in the supplementary material. To further assist the reader, we have included a statement in the main text and a table in the supplementary material on interpreting Bayes factors.

'Related line 106/108/109, I like how the models are laid out clearly in the methods. I think it would be nice, for reference, to have a table in the supplement showing the results of each of these three regressions including p-values for the 3 different climate variables, showing with and without controls for province (i.e. removing mean province outbreaks size as is done in line 183).'

Tables have been added to the supplement showing the results of each of the fitted model, including the median and 95% credible intervals of each predictor, β_i , and Bayes factors for each model fit.

'Did the authors try to control for common temporal trends across locations? This is often done with year dummies e.g. in a classic panel regression framework where both spatial and temporal controls are included?'

We did not try to control for common temporal trends across locations. Our aim was to find a consistent data-driven pattern between climate and epidemiological dynamics of the plague. While there is clearly merit in the approach you suggest - looking for common yearly trends - this did not match with the ethos of the paper. In addition, given that the timing of outbreaks was largely consistent between years (as shown in supplementary figures 2 and 5), we do not believe adding a yearly component would greatly improve the fit of our model.

‘Fig4, for lower end of humidity range 40, e.g. in the top two plots, there seems to be no observations (little circles). Is the quadratic only fitting to the upper zero points? i.e. is it really a quadratic fit or would a gam/spline etc work better? Or are there points here that are not visible?’

Thank-you for bringing this to our attention. At the lower end of the humidity range, there should be a low number of observations visible that did not report outbreaks, if outbreak threshold size was defined to be 0 or 1 deaths per 100,000 individuals. The quadratic is fitting both to these zero-observations in low humidity and those in the high humidity range. We have increased the opacity of the observation points so that they show more clearly on the figure.

‘Fig4, I think the temperature and precipitation version of this plot should be shown in the supplement, even if not significant.’

These figures have now been included in the supplement.

‘Line 105. Did you consider combined models e.g. temperature and precipitation together? i.e. $\beta_1 T + \beta_2 T^2 + \beta_3 P + \beta_4 P^2$. I imagine it might have similar explanatory power to RH.’

We did fit models of different combinations of temperature and rainfall, as we also believed that this could mimic humidity in some manner. However, each of these combination-models produced similar results to models of temperature and precipitation alone. We believe that this is due to the non-linear and time-lagged effects that exist between the three climate measurements.

‘I think the authors need to discuss the lack of mechanistic understanding as a caveat. I appreciate that the authors cannot easily unpack mechanisms, and the plague system is particularly complex. However, given the authors are only using statistical approaches and not using a mechanistic model, the limitations of this approach, in terms of lack of mechanistic understanding, should be discussed. Without a clear understanding of the mechanism linking RH to plague incidence, it is hard to say whether the approximation will hold in other countries still experiencing plague outbreaks.’

Thank-you for raising this limitation of our study. We now note in the discussion that the lack of mechanistic understanding of plague limits our ability to directly apply our findings to modern outbreak settings.